# Long-Term Nutritional Deficits and Growth Patterns in Children with Congenital Zika Virus Syndrome: Evidence from a Brazilian Cohort

**DOI:** 10.3390/v17091239

**Published:** 2025-09-14

**Authors:** Carolina Santos Souza Tavares, Raquel Souza Marques, Janiele de Sá Ferreira, Marcela Barros Barbosa de Oliveira, Monique Carla da Silva Reis, Paulo Ricardo Martins-Filho

**Affiliations:** 1Investigative Pathology Laboratory, Federal University of Sergipe, Aracaju 49060-100, SE, Brazil; enfcarol_souza@hotmail.com (C.S.S.T.); raquel.smarques@outlook.com (R.S.M.); janielesa08@gmail.com (J.d.S.F.); marcela.nutriufs@yahoo.com.br (M.B.B.d.O.); 2Department of Occupational Therapy, Alagoas State University of Health Sciences, Maceio 57010-382, AL, Brazil; moniquecsto@gmail.com

**Keywords:** Zika virus infection, Congenital Zika Syndrome, growth and development, nutritional status

## Abstract

Children with Congenital Zika Virus Syndrome (CZVS) experience severe neurological and nutritional impairments. Although immediate clinical consequences are well-documented, long-term anthropometric and nutritional outcomes remain poorly understood. This study assessed longitudinal anthropometric and nutritional outcomes in children affected by CZVS. A cohort of 38 children aged ≥ 5 years diagnosed with CZVS was followed at a reference center in Northeast Brazil. Anthropometric measures (weight, height, BMI, head circumference) were collected using standardized methods, including digital scales and anthropometric tape measures. Growth was analyzed using WHO Anthro and WHO Anthro Plus software (version 3.2.2). Dietary intake was evaluated through two 24 h recalls and analyzed with NutWIN 2.5 software. Nutritional status was classified using WHO growth standards, and associations between dietary intake and BMI were statistically examined. Children showed significant linear growth improvement (*p* = 0.007) without corresponding weight gain, leading to worsening BMI classifications (*p* = 0.017). Dietary evaluations revealed limited dietary diversity, frequent intake of ultra-processed foods, inadequate fruit consumption, and widespread insufficiencies in caloric and micronutrient intake (zinc, calcium, iron, vitamin D). Low carbohydrate intake was significantly associated with inadequate BMI (*p* = 0.030). Multidisciplinary nutritional interventions addressing medical, dietary, educational, and socioeconomic factors are essential for improving health outcomes in children with CZVS.

## 1. Introduction

Congenital Zika Virus Syndrome (CZVS) is a complex neurological disorder resulting from prenatal exposure to Zika Virus (ZIKV), a neurotropic flavivirus that emerged as a public health threat during the 2015–2016 epidemic in the Americas. The congenital phenotype is associated with maternal infection predominantly during the first and second trimesters of pregnancy [1,2], a critical period for fetal neurodevelopment [3]. CZVS has imposed long-term demands on health systems, particularly in resource-limited settings, where affected children require ongoing multidisciplinary care due to severe neurological and functional impairments [4].

Although microcephaly is the hallmark of CZVS, the syndrome encompasses a broad spectrum of clinical features, including subcortical calcifications, ocular abnormalities, early-onset hypertonia, brainstem dysfunction, and arthrogryposis. These manifestations collectively impair neuropsychomotor development, communication, cognition, and functional autonomy [5,6,7,8,9]. In addition, recent evidence indicates that neurodevelopmental alterations may also occur in normocephalic children with antenatal ZIKV exposure, including language and motor delays, reduced amygdala volumes, and lower cognitive performance at school age [10,11,12].

Beyond neurological impairments, CZVS affects gastrointestinal function, causing recurrent vomiting, gastric dysmotility, and food intolerance. These symptoms, attributed to dysfunctions in the enteric and autonomic nervous systems, significantly compromise nutritional intake and overall health [13]. Neurogenic dysphagia is particularly prevalent in this population, impairing the coordination between swallowing and breathing and affecting both nutritive and non-nutritive suckling [14]. Moreover, motor impairments associated with CZVS further exacerbate feeding difficulties by compromising postural control and swallowing mechanics. These challenges impose additional burdens on caregivers [15] and frequently result in the adoption of diets based on soft or pureed foods, which are often rich in sugars and ultra-processed components [16,17,18].

While the early clinical and nutritional impacts of CZVS are well-documented [16,19,20,21], evidence regarding its long-term effects on anthropometric development and nutritional status remains scarce. The lack of longitudinal data hinders the formulation of effective, evidence-based health and nutrition strategies. This study investigated the long-term anthropometric and nutritional outcomes in children with CZVS, providing empirical evidence to support clinical decision-making tailored to this high-risk population.

## 2. Materials and Methods

### 2.1. Study Design and Setting

This longitudinal cohort study was conducted at the University Hospital of Sergipe (HU-UFS), a regional referral center for CZVS care in Northeast Brazil. Children diagnosed with CZVS were followed from birth through early childhood. Anthropometric data collected at birth were compared with current measurements (≥5 years of age) to assess growth trajectories and changes in nutritional status. Dietary intake was also evaluated and compared between children stratified by body mass index (BMI) as adequate or inadequate for age. The study was approved by the Research Ethics Committee of the Federal University of Sergipe (approval no. 5.287.146), with informed caregiver consent.

### 2.2. Eligibility Criteria

Inclusion criteria were: (1) diagnosis of CZVS at birth according to Brazilian Ministry of Health criteria [22]; (2) availability of complete anthropometric records at birth (height, weight, and head circumference) in the Child Health Booklet (CHB); and (3) current age ≥ 5 years. These criteria enabled paired analysis of anthropometric changes over time. Children whose biological mother was not present at follow-up or whose primary caregiver was not the mother were excluded to ensure the reliability of historical information.

### 2.3. Study Size

Of 51 children monitored at HU-UFS, 38 fulfilled inclusion criteria with complete neonatal anthropometric documentation and were included in this study.

### 2.4. Outcomes and Measures

Structured interviews with mothers were conducted to obtain sociodemographic data (maternal age, area of residence, marital status, socioeconomic status, employment, and receipt of government assistance), obstetric history (pregnancy planning, prenatal care, type of delivery), neonatal data (child’s sex, Apgar scores, birth anthropometry, perinatal complications, and neonatal hospitalizations), and dietary intake of the children. Apgar scores at 1 and 5 min were retrieved from the CHB and used to assess immediate postnatal vitality based on the standard scoring system (range: 0–10).

#### 2.4.1. Anthropometric Assessment

Anthropometric outcomes were assessed at two time points: (1) at birth, using data recorded in the CHB; and (2) at the follow-up visit (≥5 years of age), conducted during the study. Parameters included weight, height, BMI, and head circumference. Neonatal anthropometric data were extracted by a trained maternal-child health nurse, following protocols established by the Brazilian Ministry of Health.

At follow-up, weight was measured using a calibrated digital scale (G-Tech^®^, 150 kg capacity). For children unable to stand due to motor impairments, weight was obtained by subtracting the caregiver’s weight from the combined weight of caregiver and child. Height was estimated using arm length (acromion to the head of the radius), measured with a flexible anthropometric tape and calculated using Stevenson’s formula: height = (4.35 × arm length) + 21.8 [23]. Head circumference was measured around the occipital bone and supraorbital arch using an inextensible measuring tape.

Growth curves for height, weight, and BMI were generated using WHO Anthro (0–60 months) and WHO Anthro Plus (5–19 years) software. Nutritional status for each parameter was classified as adequate or inadequate for age [24] (Table 1). Transitions between the two time points were categorized as: remained adequate, improved, remained inadequate, or worsened, allowing for a longitudinal assessment of individual growth trajectories. Severe microcephaly was defined as head circumference at birth below three standard deviations for gestational age and sex, according to WHO standards [25].

#### 2.4.2. Dietary Assessment

Dietary intake was assessed through two non-consecutive 24 h dietary recalls, reported by caregivers and referencing the day prior to each interview [26]. Nutritional composition was analyzed using the NutWIN 2.5 software (Federal University of São Paulo, São Paulo, Brazil), based on the Brazilian Food Composition Table. Estimates included total energy intake, macronutrients (proteins, carbohydrates, and lipids), and selected micronutrients (zinc, calcium, iron, and vitamin D). Nutritional adequacy was assessed using the Culley formula for energy requirements [27], the Acceptable Macronutrient Distribution Range (AMDR) for macronutrients [28], and the Recommended Dietary Allowances (RDA) for micronutrients [29].

### 2.5. Statistical Analysis

Descriptive statistics were used to summarize clinical, sociodemographic, dietary, and anthropometric data. Continuous variables were reported as medians and interquartile ranges (IQR), and categorical variables as absolute and relative frequencies. Longitudinal changes in anthropometric classification for height, weight, and BMI between birth and early childhood were analyzed using the McNemar test, based on dichotomous classifications (adequate vs. inadequate). Transitions in classification were categorized as: remained adequate, improved, remained inadequate, or worsened. To explore associations between current BMI status and dietary variables, inferential analyses were conducted. The Mann–Whitney test was used for continuous variables, and Fisher’s exact test for categorical variables. Statistical significance was set at *p* < 0.05 (two-tailed). Analyses were performed using R software (version 3.5.3; R Foundation for Statistical Computing).

## 3. Results

### 3.1. General Characteristics of the Participants

#### 3.1.1. Maternal Characteristics

The median age of mothers was 31 years. Most resided in urban areas (60.5%), were in stable relationships (78.9%), and unemployed (78.9%). Government financial assistance was common (92.1%), and nearly all families (94.7%) had a monthly income ranging between 1 and 3 minimum wages. Prenatal care attendance was universal, with 55.3% of mothers undergoing vaginal delivery (Table 2).

#### 3.1.2. Child Characteristics

At birth, children had a median head circumference of 29 cm and severe microcephaly was prevalent in 81.6% of cases. Slightly over half the children were male (52.6%), and the median age at assessment was 6.3 years, with a median head circumference of 44.5 cm. Frequent complications included seizures (84.2%), dysphagia (60.5%), ophthalmological disorders (42.1%), arthrogryposis (42.1%), and hypertonia (39.5%) (Table 3).

### 3.2. Anthropometric Outcomes

Longitudinal analysis demonstrated a linear growth improvement from birth through early childhood, with an increased proportion of children reaching appropriate stature for age (*p* = 0.007). Conversely, weight gain was stable, with no significant changes (*p* = 0.238). Particularly, BMI showed a significant decline over time (*p* = 0.017), with the proportion of children having an age-appropriate BMI decreasing from 73.7% at birth to 42.1% by the end of early childhood; at this stage, 77.3% of children with inadequate BMI were classified as underweight and 22.7% as overweight or obese (Table 4). Growth trajectories relative to WHO standards are visually represented in Figure 1a–c.

### 3.3. Dietary Intake and Association with BMI

The median daily meal frequency was 5, primarily administered orally (78.9%) with mushy food consistency (57.8%). Median cereal intake was 22.5%, and fruit intake was 25%. Consumption of ultra-processed foods was frequent (57.8%), while only 34.2% of children achieved minimum dietary diversity. No significant differences in dietary patterns were identified between children with adequate or inadequate BMI (Table 5).

Macronutrient analysis showed a significant association between inadequate carbohydrate intake and inadequate BMI (*p* = 0.030). No associations were observed between protein or lipid intake and BMI. Caloric intake was inadequate for 84.2% of the children, and more than 90% presented low intake of essential micronutrients, including zinc, calcium, iron, and vitamin D, without significant associations with BMI status (Table 6).

## 4. Discussion

The long-term consequences of CZVS on child growth and nutrition remain insufficiently characterized, despite the well-established neurological impairments associated with the condition. In this cohort, a distinct anthropometric pattern was observed—characterized by significant gains in height over time without proportional increases in weight or BMI. This disproportion suggests underlying compensatory biological mechanisms, potentially involving endocrine or metabolic adaptations commonly seen in chronic conditions or in contexts of sustained nutritional inadequacy [30]. Such findings highlight the complexity of growth trajectories in children with CZVS and raise concerns about the adequacy of their overall nutritional status, even when linear growth appears preserved.

The discrepancy between improved linear growth and stagnation in weight underscores important limitations in the scope and effectiveness of current multidisciplinary care. While stature gains may be attributed to sustained medical follow-up and rehabilitation, the persistence of suboptimal weight and BMI suggests that nutritional components of care are either insufficiently addressed or inadequately tailored to the needs of this population. Moreover, structural inequities, including poverty, unemployment, food insecurity, and restricted access to specialized nutritional resources, likely contribute to persistent dietary inadequacies. These factors not only influence food availability and quality but also shape feeding practices and caregiver decision-making, compounding the biological and developmental vulnerabilities associated with CZVS [17,31,32].

Dietary assessments revealed marked nutritional inadequacies, including limited dietary diversity, high consumption of ultra-processed foods, and low fruit intake. These patterns are consistent with food insecurity and structural poverty, conditions linked to adverse nutritional and developmental outcomes in childhood. In low- and middle-income countries, socioeconomic adversity and limited early stimulation are associated with poorer child development, and integrated approaches combining caregiver psychosocial stimulation with nutritional supplementation have shown partial mitigation of these deficits [33]. In the context of CZVS, these constraints are further exacerbated by chronic caregiving demands and the restricted autonomy of affected children during feeding [16].

Mechanistically, neurological impairments in CZVS lead to oromotor dysfunction and neurogenic dysphagia, compromising safe and efficient feeding, disrupting swallow–breath coordination, and increasing aspiration risk [34,35]. Chronic anticonvulsant use may further reduce appetite, alter taste, and cause gastrointestinal adverse effects, aggravating feeding routines and nutritional adequacy [36,37]. These factors, often coupled with limited caregiver knowledge and constrained access to specialized care, justify individualized feeding plans (e.g., texture/consistency modification and alternative techniques) and specialist support to secure adequate energy and nutrient intake and prevent respiratory complications.

Similar feeding challenges are well documented in other pediatric neurodevelopmental disorders, including cerebral palsy, Rett syndrome, and neuromuscular diseases [38,39,40]. In these populations, multidisciplinary programs—early referral to speech-language therapy, tailored dietary interventions, energy-dense oral supplementation, and enteral feeding when indicated—have improved growth outcomes, reduced aspiration-related hospitalizations, and alleviated caregiver burden [41,42,43]. Integrating similar protocols into the management of children with CZVS, particularly when combined with structured caregiver training, family-centered care models, and regular reassessment of feeding abilities and dietary adequacy, may yield comparable benefits and should be considered a priority in the formulation of clinical guidelines [44,45].

This study has some limitations. Its single-center design may limit external validity, as the sample may not capture the heterogeneity of children with CZVS across diverse sociocultural and geographic contexts. In addition, the use of only two 24 h dietary recalls may not fully reflect habitual intake, increasing the potential for misclassification or recall bias. In addition, the absence of a non-CZVS control group matched on socioeconomic status prevents full isolation of syndrome-specific effects. Although our within-child longitudinal comparisons (birth vs. ≥5 years) partially mitigate confounding by time-invariant factors, residual confounding cannot be excluded. Future research should prioritize multicenter longitudinal designs with broader demographic variability, inclusion of matched controls (e.g., non-CZVS peers from comparable communities or sibling-controlled studies), and more comprehensive dietary assessment methods to improve accuracy. Randomized evaluations of caregiver-focused programs, optimized dietary protocols, and early dysphagia management are also warranted to address both biological and socioeconomic determinants of nutritional risk in this vulnerable population.

## 5. Conclusions

This study demonstrated that children with CZVS present persistent nutritional inadequacies and suboptimal anthropometric trajectories, evidenced by insufficient weight gain despite improvements in stature. The observed dietary pattern, marked by high consumption of ultra-processed foods and limited diversity, reflects the combined effects of neurological impairments and socioeconomic vulnerability. Addressing these deficits requires integrated public health approaches that combine medical care, nutritional support, educational initiatives, and social assistance, with the aim of reducing nutritional risk, supporting healthy growth, and improving long-term health outcomes.

## Figures and Tables

**Figure 1 viruses-17-01239-f001:**
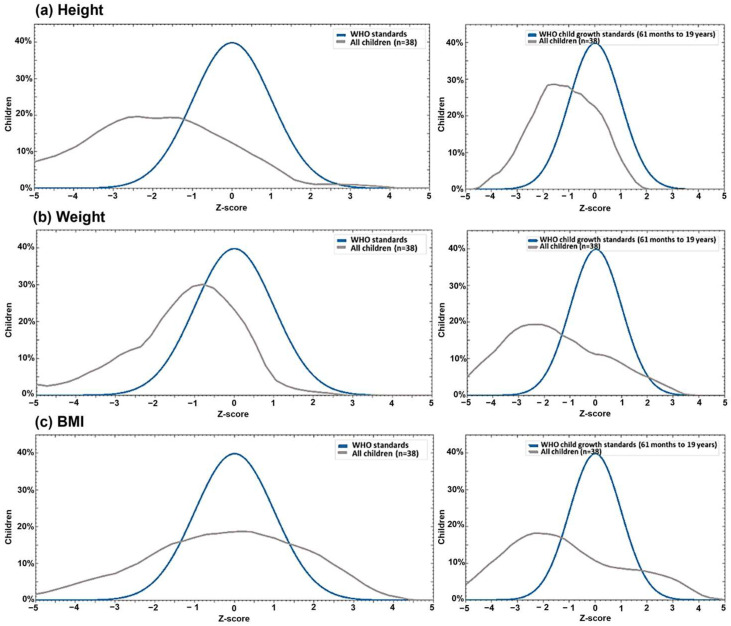
Kernel density plots of Z-scores for height (**a**), weight (**b**), and body mass index (**c**) in children with Congenital Zika Virus Syndrome compared with WHO reference standards. Left panels show measurements at birth; right panels show current measurements (≥5 years). Blue curves indicate WHO reference distributions; gray curves indicate the study cohort (*n* = 38).

**Table 1 viruses-17-01239-t001:** Cutoff points for weight, height, and body mass index.

Measures	Critical Values		Diagnosis
	Percentile	Z-Score	
Weight	>97	>+2	High weight for age.
	≥3 to ≤97	≥−2 to ≤+2	Adequate weight for age.
	≥0.1 to <3	≥−3 to <−2	Low weight for age.
	<0.1	<−3	Very low weight for age.
Height	≥3	≥−2	Adequate height for age.
	≥0.1 to <3	≥−3 to <−2	Short stature for age.
	< 0.1	<−3	Very short stature for age.
Body mass index	> 99.9	>+3	Obesity.
	>97 to ≤99.9	>+2 to ≤+3	Overweight.
	> 85 to ≤97	>+1 to ≤+2	Risk of overweight.
	≥3 to ≤85	≥−2 to ≤+1	Adequate body mass index.
	≥0.1 to <3	≥−3 to <−2	Underweight.
	<0.1	<−3	Severe underweight.

**Table 2 viruses-17-01239-t002:** Socioeconomic and obstetric characteristics of mothers of children with CZS.

Variable	*n* (%)
**Age ***	31.0 (25.0−37.0)
**Area of residence**	
Rural	15 (39.5%)
Urban	23 (60.5%)
**Marital status**	
Married/in a stable relationship	30 (78.9%)
Divorced/single	8 (21.1%)
**Employed**	
Yes	8 (21.1%)
No	30 (78.9%)
**Government benefit**	
Yes	35 (92.1%)
No	3 (7.9%)
**Monthly family income**	
Less than 1 minimum wage	2 (5.3%)
From 1 to 3 minimum wages	36 (94.7%)
**Number of births ***	2.0 (1.5−3.0)
**Live births ***	2.0 (1.0−3.0)
**Miscarriages ***	0.0 (0.0–0.0)
**Pregnancy wanted**	
Yes	16 (42.1%)
No	22 (57.9%)
**Type of delivery**	
Vaginal	21 (55.3%)
Cesarean	17 (44.7%)

* Data reported as median (Q1–Q3).

**Table 3 viruses-17-01239-t003:** Neonatal history and current data of children with CZVS.

Variable	*n* (%)
**Sex**	
Male	20 (52.6%)
Female	18 (47.4%)
**Apgar 1st minute ***	9.0 (8.0–9.0)
**Apgar 5th minute ***	10.0 (9.0–10.0)
**Head circumference at birth ***	29.0 (27.5–30.0)
**Severe microcephaly**	
Yes	31 (81.6%)
No	7 (18.4%)
**Current age**	6.3 (5.9–6.6)
**Current head circumference ***	44.5 (42.0–45.9)
**Complications**	
Arthrogryposis	16 (42.1%)
Seizures	32 (84.2%)
Dysphagia	23 (60.5%)
Ophthalmological disorders	16 (42.1%)
Hearing disorders	9 (23.7%)
Hypertonia	15 (39.5%)
Hyperreflexia	3 (7.9%)
Irritability	12 (31.6%)
Neurogenic bladder	1 (2.6%)
**Need for hospitalization**	
Yes	24 (63.2%)
No	14 (36.8%)
**Currently attends school**	
Yes	8 (21.1%)
No	30 (78.9%)

* Data reported as median (Q1–Q3).

**Table 4 viruses-17-01239-t004:** Longitudinal analysis of anthropometric measures of 38 children with CZVS.

Measures	Birth		End of Early Childhood		Diagnostic Evolution				*p*-Value ^(a)^
	Median (IQR)	Adequate Parameters for Age (%)	Median (IQR)	Adequate Parameters for Age (%)	Remained Adequate (%)	Improved (%)	Remained Inadequate (%)	Worsened (%)	
**Height (cm)**	45.0 (43.0–48.0)	16 (42.1)	108.8 (106.6–117.5)	29 (76.3)	12 (31.6)	17 (44.8)	5 (13.1)	4 (10.5)	0.007 *
**Weight (kg)**	2.7 (2.5–3.0)	28 (73.7)	15.8 (14.3–18.7)	22 (57.9)	16 (42.1)	6 (15.8)	4 (10.5)	12 (31.6)	0.238
**BMI** **(kg/m^2^)**	13.0 (11.5–14.6)	28 (73.7)	13.2 (11.8–15.0)	16 (42.1)	11 (29.0)	5 (13.1)	5 (13.1)	17 (44.8)	0.017 *

BMI, body mass index. ^(a)^ McNemar test. * *p*-values ≤ 0.05 were considered statistically significant.

**Table 5 viruses-17-01239-t005:** General dietary consumption characteristics of children with CZVS.

Variables	*n* (%) (Total = 38)	Adequate BMI (*n* = 16)	Inadequate BMI (*n* = 22)	*p*-Value ^(a)^
**Daily number of meals ***	5.0 (4.5–6.0)	5.0 (4.6–5.9)	5.3 (4.4–6.0)	0.848
**Food intake**				
Oral	30 (78.9%)	11 (68.8%)	19 (86.4%)	0.243
Enteral	8 (21.1%)	5 (31.2%)	3 (13.6%)	
**Consistency of food**				
Solid	8 (21.1%)	3 (18.8%)	5 (22.7%)	0.450
Liquid	8 (21.1%)	5 (31.2%)	3 (13.6%)	
Mushy	22 (57.8%)	8 (50.0%)	14 (63.7%)	
**% of meals with cereals ***	22.5 (2.3–37.2)	15.5 (0.0–30.1)	26.2 (11.1–37.2)	0.359
**% of meals with fruits ***	25.0 (17.1–36.4)	25.0 (17.2–37.3)	26.8 (17.1–35.6)	0.906
**Ultra-processed food consumption**				
Yes	22 (57.8%)	9 (56.3%)	13 (59.1%)	1.000
No	16 (42.2%)	7 (43.7%)	9 (40.9%)	
**Minimum dietary diversity**				
Yes	13 (34.2%)	4 (25.0%)	9 (40.9%)	0.490
No	25 (65.8%)	12 (75.0%)	13 (59.1%)	

* Data reported as median (Q1–Q3). ^(a)^ Mann–Whitney or Fisher’s exact test. BMI, body mass index.

**Table 6 viruses-17-01239-t006:** Macronutrient and micronutrient intake of children with CZVS.

Variables	*n* (%) (Total = 38)	Adequate BMI (*n* = 16)	Inadequate BMI (*n* = 22)	*p*-Value ^(a)^
**Kcal**				
Adequate	6 (15.8%)	1 (6.3%)	5 (22.7%)	0.370
Inadequate	32 (84.2%)	15 (93.7%)	17 (77.3%)	
**Macronutrients**				
**Proteins**				
Adequate	18 (47.4%)	8 (50.0%)	10 (45.5%)	1.000
Inadequate	20 (52.6%)	8 (50.0%)	12 (54.5%)	
**Carbohydrates**				
Adequate	32 (84.2%)	16 (100.0%)	16 (72.7%)	0.030 ^¥^
Inadequate	6 (15.8%)	0 (0.0%)	6 (27.3%)	
**Lipids**				
Adequate	20 (52.6%)	8 (50.0%)	12 (54.5%)	1.000
Inadequate	18 (47.4%)	8 (50.0%)	10 (45.5%)	
**Micronutrients**				
**Zinc**				
Adequate	4 (10.5%)	1 (6.3%)	3 (13.6%)	0.625
Inadequate	34 (89.5%)	15 (93.7%)	19 (86.4%)	
**Calcium**				
Adequate	1 (2.6%)	1 (6.3%)	0 (0.0%)	0.421
Inadequate	37 (97.4%)	15 (93.7%)	22 (100.0%)	
**Iron**				
Adequate	4 (10.5%)	1 (6.3%)	3 (13.6%)	0.625
Inadequate	34 (89.5%)	15 (93.7%)	19 (86.4%)	
**Vitamin D**				
Adequate	0 (0.0%)	0 (0.0%)	0 (0.0%)	1.000
Inadequate	38 (100.0%)	16 (100.0%)	22 (100.0%)	

^(a)^ Fisher’s exact test. ^¥^ *p*-values less than 0.05 were considered statistically significant.

## Data Availability

The datasets generated and/or analyzed during the current study are not publicly available due to privacy and ethical restrictions but are available from the corresponding author upon reasonable request.

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
