# Peer review of "Long-Term Nutritional Deficits and Growth Patterns in Children with Congenital Zika Virus Syndrome: Evidence from a Brazilian Cohort"

_viruses, 2025, doi:10.3390/v17091239_

Round 1
Reviewer 1 Report
Comments and Suggestions for Authors
Summary
- Authors present a descriptive longitudinal cohort study in a pediatric population diagnosed with congenital Zika virus syndrome from a single center in Brazil. The authors use standardized and validated tools to quantify the findings and report out findings.
Author Comments
- At times it is confusing to determine what groups are being compared for statistical analysis.
- Not a lot of results since this is a descriptive paper.
- Lines 165-168 seem to be the description of the section. Please delete.
- Expand discussion to include how these findings compare to other conditions with abnormal feeding patterns and dysphagia (ex. Eosinophilic esophagitis, food allergies), and what interventions have been used in those diseases.
- Table 3: please switch n and % so n is listed and % is in (). This is more aligned with clinical manuscripts (ex. JAMA). Apply this change to all tables please.
- Tables: consider including the ranges for a normal growth (ex. Average and +/- standard deviations or 3%, 50%, 97%)
- Table 5 and 6: second column title should say total (n = 38)
- Figure 1: recommend using a different color scheme due to be inclusive of colorblind readers (https://www.nature.com/articles/d41586-021-02696-z)
Author Response
Comment 1: “At times it is confusing to determine what groups are being compared for statistical analysis.”
Author Response:
We appreciate this comment and have revised the Materials and Methods section to enhance clarity regarding the analytical strategy. Specifically, we modified the opening paragraph of the section to clearly indicate that children diagnosed with CZVS were followed from birth through early childhood (≥5 years), and that anthropometric data from these two time points were compared to assess growth trajectories and changes in classification (adequate vs. inadequate for age). We also clarified that current dietary intake was compared between children stratified by BMI status (adequate vs. inadequate). Furthermore, we revised the subsection Statistical Analysis to explicitly describe the comparison groups and the statistical tests applied: (i) longitudinal changes in anthropometric classifications were analyzed using the McNemar test; and (ii) dietary characteristics were compared between BMI groups using Mann–Whitney and Fisher’s exact tests. These adjustments were made to improve methodological transparency and facilitate reader understanding of the analytical framework.
Comment 2: “Not a lot of results since this is a descriptive paper.”
Author Response:
In addition to the descriptive characterization of anthropometric and dietary profiles in children with CZVS, inferential analyses were also conducted to explore differences and changes over time. Specifically, longitudinal comparisons of nutritional status classifications (adequate vs. inadequate for age) for height, weight, and BMI between birth and early childhood (≥5 years) were performed using the McNemar test (Table 4). Furthermore, dietary intake characteristics were compared between BMI groups (adequate vs. inadequate) using the Mann–Whitney and Fisher’s exact tests (Tables 5 and 6). These analyses aimed to provide a more robust understanding of growth trajectories and nutritional adequacy in this high-risk population. To reinforce the relevance of these findings, we revised the Results section to improve clarity and highlight key statistical outcomes that may inform future clinical and public health strategies.
Comment 3: “Lines 165–168 seem to be the description of the section. Please delete.”
Author Response:
We thank the reviewer for this observation. The sentence in question has been removed to maintain conciseness and avoid redundancy with the section's title and content.
Comment 4: "Expand discussion to include how these findings compare to other conditions with abnormal feeding patterns and dysphagia (e.g., Eosinophilic esophagitis, food allergies), and what interventions have been used in those diseases."
Author Response:
We thank the reviewer for this constructive suggestion. In response, we expanded the Discussion section to contextualize our findings in relation to other pediatric conditions associated with feeding difficulties and dysphagia, particularly those involving neurodevelopmental impairment, such as cerebral palsy, Rett syndrome, and neuromuscular disorders. We also included evidence on multidisciplinary interventions commonly adopted in these conditions—including early involvement of speech-language therapists, individualized dietary modifications, nutritional supplementation, and enteral feeding support—and discussed their potential applicability to children with CZVS. These additions aim to enhance the translational value of our findings and highlight clinical strategies that may improve nutritional outcomes and reduce caregiver burden in this population.
Added references:
- Sullivan, P.B.; Juszczak, E.; Lambert, B.R.; Rose, M.; Ford-Adams, M.E.; Johnson, A. Impact of feeding problems on nutritional intake and growth: Oxford Feeding Study II. Dev. Med. Child Neurol. 2002, 44, 461-7, doi:10.1017/s0012162201002365.
- Oddy, W.H.; Webb, K.G.; Baikie, G.; Thompson, S.M.; Reilly, S.; Fyfe, S.D.; Young, D.; Anderson, A.M.; Leonard, H. Feeding experiences and growth status in a Rett syndrome population. J. Pediatr. Gastroenterol. Nutr. 2007, 45, 582-90, doi:10.1097/MPG.0b013e318073cbf7.
- Dipasquale, V.; Morello, R.; Romano, C. Gastrointestinal and nutritional care in pediatric neuromuscular disorders. World J. Clin. Pediatr. 2023, 12,197-204, doi:10.5409/wjcp.v12.i4.197.
- Chou, E.; Lindeback, R.; Sampaio, H.; Farrar, M.A. Nutritional practices in pediatric patients with neuromuscular disorders. Nutr. Rev. 2020, 78, 857-865, doi: 10.1093/nutrit/nuz109.
- Pinto, C.; Borrego, R.; Eiró-Gomes, M.; Casimiro, I.; Raposo, A.; Folha, T.; Virella, D.; Moreira, A.C. Embracing the Nutritional Assessment in Cerebral Palsy: A Toolkit for Healthcare Professionals for Daily Practice. Nutrients. 2022, 14, 1180, doi: 10.3390/nu14061180.
- Quitadamo, P.; Thapar, N.; Staiano, A.; Borrelli, O. Gastrointestinal and nutritional problems in neurologically impaired children. Eur. J. Paediatr. Neurol. 2016, 20, 810-815, doi:10.1016/j.ejpn.2016.05.019.
- Frawley, H.E.; Andrews, S.M.; Wheeler, A.C.; Nobrega, L.L.; Firmino, R.C.B.; da Silva, C.M.; Bezerra, P.; Ventura, C.V.; Cavalcanti, A.; Williams, J.; et al. Feeding practices and weight status of children with congenital Zika syndrome: A longitudinal study in Brazil. J. Pediatr. Gastroenterol. Nutr. 2024, 79, 679-687, doi: 10.1002/jpn3.12304.
- Smythe, T.; Matos, M.; Reis, J.; Duttine, A.; Ferrite, S.; Kuper, H. Mothers as facilitators for a parent group intervention for children with Congenital Zika Syndrome: Qualitative findings from a feasibility study in Brazil. PLoS One. 2020, 15, e0238850, doi: 10.1371/journal.pone.0238850.
Comment 5: Table 3: please switch n and % so n is listed and % is in (). This is more aligned with clinical manuscripts (ex. JAMA). Apply this change to all tables please.
Author Response:
We appreciate the suggestion and have revised Table 3—as well as all other applicable tables in the manuscript—to follow the recommended format.
Comment 6: Tables: consider including the ranges for a normal growth (e.g., average and ± standard deviations or 3%, 50%, 97%).
Author Response:
Dear reviewer, we opted not to include reference growth ranges (e.g., means, standard deviations, or percentiles) directly in the results tables to avoid visual and informational overload. Instead, the cutoff points used to classify weight, height, and body mass index (BMI) as adequate or inadequate for age—based on WHO growth standards—are detailed in Table 1. This approach ensures methodological transparency while maintaining the clarity and readability of the main results tables.
Comment 7: Table 5 and 6: second column title should say total (n = 38).
Author Response:
We appreciate the observation. The column titles in Tables 5 and 6 have been updated to indicate “Total (n = 38)” as requested.
Comment 8: Figure 1: recommend using a different color scheme due to be inclusive of colorblind readers (https://www.nature.com/articles/d41586-021-02696-z)
Author Response:
We thank the reviewer for this important suggestion. Figure 1 has been improved to enhance clarity and accessibility. Font size and overall readability have been optimized, and the color scheme has been updated to be colorblind-friendly, with growth curves now represented in blue and gray, following recommended accessibility guidelines.
Reviewer 2 Report
Comments and Suggestions for Authors
In this study, Tavares and Colleagues assessed the long-term nutritional and growth outcomes in children with Congenital Zika Virus Syndrome (CZVS). A cohort of 38 children (age: 6.3 years old (5.9-6.6) diagnosed with CZVS at a reference center in Northeast Brazil. Children showed significant linear growth improvement (p=0.007) without corresponding weight gain, leading to worsening BMI rate (p=0.017). Dietary evaluations revealed limited dietary diversity, frequent intake of ultra-processed foods, inadequate fruit consumption, and widespread insufficiencies in caloric and micronutrient intake (zinc, calcium, iron, vitamin D). Low carbohydrate intake was significantly associated with inadequate BMI (p=0.030). The Authors concluded that multidisciplinary nutritional interventions addressing medical, dietary, educational, and socioeconomic factors are essential for improving health outcomes in children with CZVS. The data collected were exhaustively analyzed and they adequately support the conclusion advanced by the Author. I have to suggest the following minor changes:
- I would like to revise Table 1. You could split the column “critical values” in 2 columns, Percentile and Z-score, and indicate the corresponding values under each column.
- I would remove the row “Prenatal” in Table 2 and report these data in the text only.
- I would use borders for the 4 columns under “Diagnostic Evolution” in Table 4.
- I would describe the Apgar method in Material and Methods section.
Author Response
Comment 1: In this study, Tavares and Colleagues assessed the long-term nutritional and growth outcomes in children with Congenital Zika Virus Syndrome (CZVS). A cohort of 38 children (age: 6.3 years old (5.9-6.6) diagnosed with CZVS at a reference center in Northeast Brazil. Children showed significant linear growth improvement (p=0.007) without corresponding weight gain, leading to worsening BMI rate (p=0.017). Dietary evaluations revealed limited dietary diversity, frequent intake of ultra-processed foods, inadequate fruit consumption, and widespread insufficiencies in caloric and micronutrient intake (zinc, calcium, iron, vitamin D). Low carbohydrate intake was significantly associated with inadequate BMI (p=0.030). The Authors concluded that multidisciplinary nutritional interventions addressing medical, dietary, educational, and socioeconomic factors are essential for improving health outcomes in children with CZVS. The data collected were exhaustively analyzed and they adequately support the conclusion advanced by the Author. I have to suggest the following minor changes:
I would like to revise Table 1. You could split the column “critical values” in 2 columns, Percentile and Z-score, and indicate the corresponding values under each column.
Author Response:
We sincerely thank the reviewer for the positive and encouraging comments regarding our study and its relevance. As suggested, Table 1 has been reformatted to improve clarity. The column “Critical values” was split into two separate columns (“Percentile” and “Z-score”), with the corresponding values presented under each heading. We believe this adjustment enhances the table’s readability and facilitates interpretation of the anthropometric cutoffs used in the analysis.
Comment 2: I would remove the row “Prenatal” in Table 2 and report these data in the text only.
Author Response:
The row “Prenatal” in Table 2 has been removed, and the corresponding information is now reported in the text only.
Comment 3: I would use borders for the 4 columns under “Diagnostic Evolution” in Table 4.
Author Response:
Borders have been added to the four columns under “Diagnostic Evolution” in Table 4 to enhance visual separation and improve table readability.
Comment 4: I would describe the Apgar method in Material and Methods section.
Author Response:
We appreciate the reviewer’s suggestion. The Materials and Methods section has been updated to include a description of the Apgar method as follows: “Apgar scores at 1 and 5 minutes were retrieved from the CHB and used to assess immediate postnatal vitality based on the standard scoring system (range: 0–10).”
Reviewer 3 Report
Comments and Suggestions for Authors
This manuscript addresses an important topic by tracking the anthropometric data and nutritional intake of 38 pediatric patients diagnosed with congenital Zika syndrome. The research question is relevant and timely. However, the study has a major limitation that significantly affects its ability to draw conclusions about growth patterns specifically attributable to congenital Zika syndrome.
According to the data presented, both anthropometric development and nutritional profiles appear to be strongly influenced by a key confounding factor: 35 out of the 38 children included in the study come from socioeconomically disadvantaged backgrounds. This inherently involves overconsumption of ultra-processed foods and limited dietary diversity, which are well-known to impact child growth and nutrition independently of any congenital condition.
The comparison with WHO growth standards (Figure 1) does not adequately address this limitation. To isolate and better understand the specific effects on growth of children with congenital Zika syndrome it would be necessary to include a control group of children without the syndrome but with comparable socioeconomic conditions. This would allow for a clearer differentiation between the effects of socioeconomic status and those attributable directly to the syndrome itself.
As it stands, the study primarily demonstrates patterns already commonly observed among children from low-income households, rather than isolating syndrome-specific growth outcomes.
As minor comments, the legend of Figure 1 is unclear, and the difference between the left and right panels should be explained more explicitly.
Author Response
Comment 1: This manuscript addresses an important topic by tracking the anthropometric data and nutritional intake of 38 pediatric patients diagnosed with congenital Zika syndrome. The research question is relevant and timely. However, the study has a major limitation that significantly affects its ability to draw conclusions about growth patterns specifically attributable to congenital Zika syndrome.
According to the data presented, both anthropometric development and nutritional profiles appear to be strongly influenced by a key confounding factor: 35 out of the 38 children included in the study come from socioeconomically disadvantaged backgrounds. This inherently involves overconsumption of ultra-processed foods and limited dietary diversity, which are well-known to impact child growth and nutrition independently of any congenital condition.
The comparison with WHO growth standards (Figure 1) does not adequately address this limitation. To isolate and better understand the specific effects on growth of children with congenital Zika syndrome it would be necessary to include a control group of children without the syndrome but with comparable socioeconomic conditions. This would allow for a clearer differentiation between the effects of socioeconomic status and those attributable directly to the syndrome itself. As it stands, the study primarily demonstrates patterns already commonly observed among children from low-income households, rather than isolating syndrome-specific growth outcomes.
Author Response:
We agree that socioeconomic disadvantage is a major determinant of child growth and diet and have made this explicit in the revised Discussion and Limitations. Our cohort, however, is socioeconomically homogeneous (predominantly low-income), which limits internal variability for SES contrasts and precluded inclusion of a matched non-CZVS control group in this clinical follow-up. To strengthen inference within these constraints, we analyzed within-child longitudinal changes (birth vs. ≥5 years), which partially control for time-invariant confounding and revealed a discordant trajectory—linear growth improvement (p=0.007) alongside BMI deterioration (p=0.017). This pattern is consistent with disease-related feeding impairment described in CZVS (high dysphagia burden, breastfeeding/feeding difficulties) and in other neurogenic conditions, where oropharyngeal dysfunction, postural/motor limitations, and medication effects contribute to inadequate intake independent of SES (Sullivan et al.; 2022; Oddy et al.; 2007; Dipasquale et al.; 2023). We have added text highlighting this mechanistic plausibility and explicitly acknowledge that SES remains a residual confounder. We also outline, as a priority for future work, matched-cohort designs (e.g., non-CZVS peers from comparable communities or sibling-controlled studies) to better isolate syndrome-specific effects. Our objective here is to provide real-world, long-term trajectories for a highly vulnerable population to inform clinical and public-health action, not to claim causal attribution to CZVS alone.
Added references:
- Sullivan, P.B.; Juszczak, E.; Lambert, B.R.; Rose, M.; Ford-Adams, M.E.; Johnson, A. Impact of feeding problems on nutritional intake and growth: Oxford Feeding Study II. Dev. Med. Child Neurol. 2002, 44, 461-7, doi:10.1017/s0012162201002365.
- Oddy, W.H.; Webb, K.G.; Baikie, G.; Thompson, S.M.; Reilly, S.; Fyfe, S.D.; Young, D.; Anderson, A.M.; Leonard, H. Feeding experiences and growth status in a Rett syndrome population. J. Pediatr. Gastroenterol. Nutr. 2007, 45, 582-90, doi:10.1097/MPG.0b013e318073cbf7.
- Dipasquale, V.; Morello, R.; Romano, C. Gastrointestinal and nutritional care in pediatric neuromuscular disorders. World J. Clin. Pediatr. 2023, 12,197-204, doi:10.5409/wjcp.v12.i4.197.
Additionally, we took this opportunity to improve the Introduction by providing a more comprehensive background on the relationship between Zika virus infection and microcephaly, to refine the Methods section for greater clarity and reproducibility, and to enhance the Discussion with contextual interpretation of our findings in light of the existing literature.
Added references:
- Mendes, A.K.T.; Ribeiro, M.R.C.; Lamy-Filho, F.; Amaral, G.C.; Borges, M.C.R.; Costa, L.C.; Cavalcante, T.B.; Batista, R.F.L.; Sousa, P.S.; Silva, A.A.M. Congenital Zika syndrome: association between the gestational trimester of maternal infection, se-verity of brain computed tomography findings and microcephaly at birth. Rev. Inst. Med. Trop. Sao Paulo. 2020, 62, e56, doi:10.1590/s1678-9946202062056.
- Schuller-Faccini, L.; Ribeiro, E.M.; Feitosa, I.M.L.; Horovitz, D.D.G.; Cavalcanti, D.P.; Pessoa, A.; Doriqui, M.J.R.; Neri, J.I.; Pina Neto, J.M.; Wanderley, H.Y.C.; et al. Possible Association Between Zika Virus Infection and Microcephaly - Brazil, 2015. MMWR Morb. Mortal Wkly. Rep. 2016, 65, 59-62, doi:10.15585/mmwr.mm6503e2.
- Leibovitz, Z.; Leran-Sagie, T.; Haddad, L. Fetal Brain Development: Regulating Processes and Related Malformations. Life (Basel). 2022, 12, 809, doi:10.3390/life12060809.
- Martins-Filho, P.R.; Lima, T.R.C.M.; Ferreira, J.S.; Souza, L.S.; Guerra, C.B.M.C.; Santos-Júnior, L.C.; Marques, R.S.; Tavares, C.S.S.; Araújo, B.C.L. Integrated Functional Care for Children with Congenital Zika Syndrome: Addressing Orofacial, Speech-Language, and Nutritional Needs. Oral Dis. 2025, Aug 4, doi:10.1111/odi.70058.
- Marbán-Castro, E.; Guillamet, L.J.V.; Pantoja, P.E.; Casellas, A.; Maxwell, L.; Mulkey, S.B.; Menéndez, C.; Bardají, A. Neu-rodevelopment in Normocephalic Children Exposed to Zika Virus in Utero with No Observable Defects at Birth: A Systematic Review with Meta-Analysis. Int. J. Environ. Res. Public Health. 2022, 19, 7319, doi:10.3390/ijerph19127319.
- Ghosh, S.; Salan, T.; Riotti, J.; Ramachandran, A.; Gonzalez, I.A.; Bandstra, E.S.; Reyes, F.L; Andreansky, S.S.; Govind, V.; Saigal, G. Brain MRI segmentation of Zika-Exposed normocephalic infants shows smaller amygdala volumes. PLoS One. 2023, 18, e0289227, doi:10.1371/journal.pone.0289227.
- Mulkley, S.B.; Andringa-Seed, R.; Corn, E.; Williams, M.E.; Arroyave-Wessel, M.; Podolsky, R.H.; Peyton, C.; Msall, M.E.; Cure, C.; Berl, M.M. School-age child neurodevelopment following antenatal Zika virus exposure. Pediatr. Res. 2025, Mar 19, 10.1038/s41390-025-03981-7. doi:10.1038/s41390-025-03981-7.
- Antoniou, E.; Andronikidi, P.E.; Eskitzis, P.; Iliadou, M.; Palaska, E.; Tzitiridou-Chatzopoulou, M.; Rigas, N.; Orovou, E. Congenital Zika Syndrome and Disabilities of Feeding and Breastfeeding in Early Childhood: A Systematic Review. Viruses. 2023, 15, 601, doi:10.3390/v15030601.
- Soares, F.; Abranches, A.D.; Villela, L.; Lara, S.; Araújo, D.; Nehab, S.; Silva, L.; Amaral, Y.; Junior, S.C.G.; Pone, S.; et al. Zika virus infection in pregnancy and infant growth, body composition in the first three months of life: a cohort study. Sci. Rep., 2019, 9, 19198, doi:10.1038/s41598-019-55598-6.
- Arrais, N.M.R.; Maia, C.R.S.; Jerônimo, S.M.B.; Neri, J.I.C.F.; Melo, A.N.; Bezerra, M.T.A.L.; Moraes-Pinto, M.I. Growth and Survival of a Cohort of Congenital Zika Virus Syndrome Children Born with Microcephaly and Children Who Developed with Microcephaly After Birth. Pediatr. Infect. Dis. J., 2025, 44, 465-472, doi:10.1097/INF.0000000000004706.
- Chou, E.; Lindeback, R.; Sampaio, H.; Farrar, M.A. Nutritional practices in pediatric patients with neuromuscular disorders. Nutr. Rev. 2020, 78, 857-865, doi: 10.1093/nutrit/nuz109.
- Pinto, C.; Borrego, R.; Eiró-Gomes, M.; Casimiro, I.; Raposo, A.; Folha, T.; Virella, D.; Moreira, A.C. Embracing the Nutritional Assessment in Cerebral Palsy: A Toolkit for Healthcare Professionals for Daily Practice. Nutrients. 2022, 14, 1180, doi: 10.3390/nu14061180.
- Quitadamo, P.; Thapar, N.; Staiano, A.; Borrelli, O. Gastrointestinal and nutritional problems in neurologically impaired children. Eur. J. Paediatr. Neurol. 2016, 20, 810-815, doi:10.1016/j.ejpn.2016.05.019.
- Frawley, H.E.; Andrews, S.M.; Wheeler, A.C.; Nobrega, L.L.; Firmino, R.C.B.; da Silva, C.M.; Bezerra, P.; Ventura, C.V.; Cavalcanti, A.; Williams, J.; et al. Feeding practices and weight status of children with congenital Zika syndrome: A longitudinal study in Brazil. J. Pediatr. Gastroenterol. Nutr. 2024, 79, 679-687, doi: 10.1002/jpn3.12304.
- Smythe, T.; Matos, M.; Reis, J.; Duttine, A.; Ferrite, S.; Kuper, H. Mothers as facilitators for a parent group intervention for children with Congenital Zika Syndrome: Qualitative findings from a feasibility study in Brazil. PLoS One. 2020, 15, e0238850, doi: 10.1371/journal.pone.0238850.
Comment 2: As minor comments, the legend of Figure 1 is unclear, and the difference between the left and right panels should be explained more explicitly.
Author Response:
We thank the reviewer for this observation. Following the comments from Reviewer 1, Figure 1 has already been improved in terms of clarity, font size, and color palette, now using a colorblind-friendly scheme (blue for WHO standards and gray for all children with CZVS). To further address the present comment, we revised the figure legend to clarify the distinction between the left and right panels:
Revised legend:
“Figure 1. Kernel density plots of Z-scores for height (a), weight (b), and body mass index (c) in children with Congenital Zika Virus Syndrome compared with WHO reference standards. Left panels show measurements at birth; right panels show current measurements (≥5 years). Blue curves indicate WHO reference distributions; gray curves indicate the study cohort (n = 38).”
We believe this modification improves the interpretability of the figure while maintaining visual clarity.
Reviewer 4 Report
Comments and Suggestions for Authors
This is an interesting study. However, some significant references are missing. Example:
Zika Virus and the Risk of Developing Microcephaly in Infants: A Systematic Review. Antoniou E, et al. Int J Environ Res Public Health. 2020 May 27;17(11):3806. doi: 10.3390/ijerph17113806.
Author Response
Comment 1: This is an interesting study. However, some significant references are missing. Example: Zika Virus and the Risk of Developing Microcephaly in Infants: A Systematic Review. Antoniou E, et al. Int J Environ Res Public Health. 2020 May 27;17(11):3806. doi: 10.3390/ijerph17113806.
Author Response:
We thank the reviewer for the constructive feedback and for suggesting the reference by Antoniou et al. (2020). This citation has been incorporated into the manuscript as recommended. Additionally, we took this opportunity to improve the Introduction by providing a more comprehensive background on the relationship between Zika virus infection and microcephaly, to refine the Methods section for greater clarity and reproducibility, and to enhance the Discussion with contextual interpretation of our findings in light of the existing literature.
Added references:
- Mendes, A.K.T.; Ribeiro, M.R.C.; Lamy-Filho, F.; Amaral, G.C.; Borges, M.C.R.; Costa, L.C.; Cavalcante, T.B.; Batista, R.F.L.; Sousa, P.S.; Silva, A.A.M. Congenital Zika syndrome: association between the gestational trimester of maternal infection, se-verity of brain computed tomography findings and microcephaly at birth. Rev. Inst. Med. Trop. Sao Paulo. 2020, 62, e56, doi:10.1590/s1678-9946202062056.
- Schuller-Faccini, L.; Ribeiro, E.M.; Feitosa, I.M.L.; Horovitz, D.D.G.; Cavalcanti, D.P.; Pessoa, A.; Doriqui, M.J.R.; Neri, J.I.; Pina Neto, J.M.; Wanderley, H.Y.C.; et al. Possible Association Between Zika Virus Infection and Microcephaly - Brazil, 2015. MMWR Morb. Mortal Wkly. Rep. 2016, 65, 59-62, doi:10.15585/mmwr.mm6503e2.
- Leibovitz, Z.; Leran-Sagie, T.; Haddad, L. Fetal Brain Development: Regulating Processes and Related Malformations. Life (Basel). 2022, 12, 809, doi:10.3390/life12060809.
- Martins-Filho, P.R.; Lima, T.R.C.M.; Ferreira, J.S.; Souza, L.S.; Guerra, C.B.M.C.; Santos-Júnior, L.C.; Marques, R.S.; Tavares, C.S.S.; Araújo, B.C.L. Integrated Functional Care for Children with Congenital Zika Syndrome: Addressing Orofacial, Speech-Language, and Nutritional Needs. Oral Dis. 2025, Aug 4, doi:10.1111/odi.70058.
- Marbán-Castro, E.; Guillamet, L.J.V.; Pantoja, P.E.; Casellas, A.; Maxwell, L.; Mulkey, S.B.; Menéndez, C.; Bardají, A. Neu-rodevelopment in Normocephalic Children Exposed to Zika Virus in Utero with No Observable Defects at Birth: A Systematic Review with Meta-Analysis. Int. J. Environ. Res. Public Health. 2022, 19, 7319, doi:10.3390/ijerph19127319.
- Ghosh, S.; Salan, T.; Riotti, J.; Ramachandran, A.; Gonzalez, I.A.; Bandstra, E.S.; Reyes, F.L; Andreansky, S.S.; Govind, V.; Saigal, G. Brain MRI segmentation of Zika-Exposed normocephalic infants shows smaller amygdala volumes. PLoS One. 2023, 18, e0289227, doi:10.1371/journal.pone.0289227.
- Mulkley, S.B.; Andringa-Seed, R.; Corn, E.; Williams, M.E.; Arroyave-Wessel, M.; Podolsky, R.H.; Peyton, C.; Msall, M.E.; Cure, C.; Berl, M.M. School-age child neurodevelopment following antenatal Zika virus exposure. Pediatr. Res. 2025, Mar 19, 10.1038/s41390-025-03981-7. doi:10.1038/s41390-025-03981-7.
- Antoniou, E.; Andronikidi, P.E.; Eskitzis, P.; Iliadou, M.; Palaska, E.; Tzitiridou-Chatzopoulou, M.; Rigas, N.; Orovou, E. Congenital Zika Syndrome and Disabilities of Feeding and Breastfeeding in Early Childhood: A Systematic Review. Viruses. 2023, 15, 601, doi:10.3390/v15030601.
- Soares, F.; Abranches, A.D.; Villela, L.; Lara, S.; Araújo, D.; Nehab, S.; Silva, L.; Amaral, Y.; Junior, S.C.G.; Pone, S.; et al. Zika virus infection in pregnancy and infant growth, body composition in the first three months of life: a cohort study. Sci. Rep., 2019, 9, 19198, doi:10.1038/s41598-019-55598-6.
- Arrais, N.M.R.; Maia, C.R.S.; Jerônimo, S.M.B.; Neri, J.I.C.F.; Melo, A.N.; Bezerra, M.T.A.L.; Moraes-Pinto, M.I. Growth and Survival of a Cohort of Congenital Zika Virus Syndrome Children Born with Microcephaly and Children Who Developed with Microcephaly After Birth. Pediatr. Infect. Dis. J., 2025, 44, 465-472, doi:10.1097/INF.0000000000004706.
- Sullivan, P.B.; Juszczak, E.; Lambert, B.R.; Rose, M.; Ford-Adams, M.E.; Johnson, A. Impact of feeding problems on nutritional intake and growth: Oxford Feeding Study II. Dev. Med. Child Neurol. 2002, 44, 461-7, doi:10.1017/s0012162201002365.
- Oddy, W.H.; Webb, K.G.; Baikie, G.; Thompson, S.M.; Reilly, S.; Fyfe, S.D.; Young, D.; Anderson, A.M.; Leonard, H. Feeding experiences and growth status in a Rett syndrome population. J. Pediatr. Gastroenterol. Nutr. 2007, 45, 582-90, doi:10.1097/MPG.0b013e318073cbf7.
- Dipasquale, V.; Morello, R.; Romano, C. Gastrointestinal and nutritional care in pediatric neuromuscular disorders. World J. Clin. Pediatr. 2023, 12,197-204, doi:10.5409/wjcp.v12.i4.197.
- Chou, E.; Lindeback, R.; Sampaio, H.; Farrar, M.A. Nutritional practices in pediatric patients with neuromuscular disorders. Nutr. Rev. 2020, 78, 857-865, doi: 10.1093/nutrit/nuz109.
- Pinto, C.; Borrego, R.; Eiró-Gomes, M.; Casimiro, I.; Raposo, A.; Folha, T.; Virella, D.; Moreira, A.C. Embracing the Nutritional Assessment in Cerebral Palsy: A Toolkit for Healthcare Professionals for Daily Practice. Nutrients. 2022, 14, 1180, doi: 10.3390/nu14061180.
- Quitadamo, P.; Thapar, N.; Staiano, A.; Borrelli, O. Gastrointestinal and nutritional problems in neurologically impaired children. Eur. J. Paediatr. Neurol. 2016, 20, 810-815, doi:10.1016/j.ejpn.2016.05.019.
- Frawley, H.E.; Andrews, S.M.; Wheeler, A.C.; Nobrega, L.L.; Firmino, R.C.B.; da Silva, C.M.; Bezerra, P.; Ventura, C.V.; Cavalcanti, A.; Williams, J.; et al. Feeding practices and weight status of children with congenital Zika syndrome: A longitudinal study in Brazil. J. Pediatr. Gastroenterol. Nutr. 2024, 79, 679-687, doi: 10.1002/jpn3.12304.
- Smythe, T.; Matos, M.; Reis, J.; Duttine, A.; Ferrite, S.; Kuper, H. Mothers as facilitators for a parent group intervention for children with Congenital Zika Syndrome: Qualitative findings from a feasibility study in Brazil. PLoS One. 2020, 15, e0238850, doi: 10.1371/journal.pone.0238850.
Round 2
Reviewer 1 Report
Comments and Suggestions for Authors
Introduction
- Appreciate that this section is more robust with improved references.
- Authors addressed concerns about impact and importance of this manuscript.
Materials and Methods
- Authors improved description of eligibility criteria
- Addressed questions about statistics in methods section.
Results
- Authors addressed concerns about tables and they are much easier to interpret now.
- Description of characteristics correlates with tables better.
- Table 4: for the diagnostic evolution section, add (%) after each of the column titles
- Figure 1: Axes on graphs are blurry. Consider export plots from R into higher resolution PDF as they will be vector images (vs pixel images), before editing to add labels, as this may preserve the resolution better.
- Other reviewer comments were addressed.
Discussion
- Authors addressed the reviewers concerns about comparing CZVS to other pediatric conditions.
Author Response
REVIEWER 1
We sincerely thank the reviewer for the thoughtful comments, which contributed meaningfully to improving the quality of our manuscript. We have made a concerted effort to address all suggestions raised during the review process.
Reviewer’s comment (summary).
“Figure 1: Axes on graphs are blurry. Consider exporting plots from R into higher-resolution PDF (vector) before editing to add labels, as this may preserve the resolution better.”
Authors’ response.
We have replaced Figure 1 with a TIFF file at 1200 dpi, using LZW compression, generated via 3× upscaling and the application of a gentle unsharp mask (with conservative parameters). This resulted in substantially crisper and more legible axes and labels at the final print size. Importantly, no changes were made to the underlying data, plotted curves, or their interpretation—only typographic and visual refinements were applied. We believe this addresses the concern regarding figure clarity.
Reviewer’s comment (summary).
“Table 4: for the diagnostic evolution section, add (%) after each of the column titles.”
Authors’ response.
We have updated Table 4 to include “(%)” after each column title in the diagnostic evolution section, as requested.
Reviewer 3 Report
Comments and Suggestions for Authors
the authors answers to all of my comments
Author Response
We thank the reviewer for the positive feedback.